# Role of Tim17 in coupling the import motor to the translocation channel of the mitochondrial presequence translocase

Keren Demishtein-Zohary[1†], Umut Günsel[2†], Milit Marom[1], Rupa Banerjee[2], Walter Neupert[3], Abdussalam Azem[1*], Dejana Mokranjac[2*]

[1]Department of Biochemistry and Molecular Biology, The George S. Wise Faculty of Life Sciences, Tel Aviv University, Tel Aviv, Israel; [2]BMC-Physiological Chemistry, LMU Munich, Martinsried, Germany; [3]Max Planck Institute of Biochemistry, Martinsried, Germany

**Abstract** The majority of mitochondrial proteins use N-terminal presequences for targeting to mitochondria and are translocated by the presequence translocase. During translocation, proteins, threaded through the channel in the inner membrane, are handed over to the import motor at the matrix face. Tim17 is an essential, membrane-embedded subunit of the translocase; however, its function is only poorly understood. Here, we functionally dissected its four predicted transmembrane (TM) segments. Mutations in TM1 and TM2 impaired the interaction of Tim17 with Tim23, component of the translocation channel, whereas mutations in TM3 compromised binding of the import motor. We identified residues in the matrix-facing region of Tim17 involved in binding of the import motor. Our results reveal functionally distinct roles of different regions of Tim17 and suggest how they may be involved in handing over the proteins, during their translocation into mitochondria, from the channel to the import motor of the presequence translocase.

*For correspondence: azema@tauex.tau.ac.il (AA); dejana.mokranjac@med.uni-muenchen.de (DM)

[†]These authors contributed equally to this work

**Competing interests:** The authors declare that no competing interests exist.

## Introduction

Biogenesis of mitochondria requires import of over 1000 different proteins synthesized as precursor proteins in the cytosol. More than half of them carry N-terminal presequences and use the TOM and the TIM23 complexes for their translocation across the outer and inner membranes, respectively (*Bohnert et al., 2015*; *Endo et al., 2011*; *Marom et al., 2011a*; *Schulz et al., 2015*). Intermembrane space (IMS)-exposed domains of Tim50 and Tim23 serve as receptors of the TIM23 complex that recognize proteins as soon as they appear at the outlet of the TOM complex (*Marom et al., 2011b*; *Mokranjac et al., 2009*; *Schulz et al., 2011*; *Tamura et al., 2009*). In a membrane potential-dependent step, proteins are then transferred to the translocation channel in the inner membrane formed by membrane-embedded segments of Tim23 and possibly also Tim17 and/or Mgr2 (*Alder et al., 2008a*; *Dekker et al., 1997*; *Ieva et al., 2014*; *Martinez-Caballero et al., 2007*; *Truscott et al., 2001*). Complete translocation into the matrix is mediated by the import motor of the TIM23 complex, also referred to as PAM (presequence translocase-associated motor). mtHsp70 is the central component of the import motor. Its ATP-dependent action, regulated by the J and J-like proteins Tim14/Pam18 and Tim16/Pam16 and the nucleotide exchange factor Mge1, drives translocation of proteins across the inner membrane (*D'Silva et al., 2003*; *Frazier et al., 2004*; *Kang et al., 1990*; *Liu et a., 2003*; *Mokranjac et al., 2003*). Tim44 is a peripheral membrane protein that acts as an anchor to recruit the import motor to the translocation channel (*D'Silva et al., 2004*; *Kozany et al., 2004*; *Schulz and Rehling, 2014*). Through its N-terminal domain, Tim44 binds to the components

of the import motor, whereas its C-terminal domain binds to Tim17 (*Banerjee et al., 2015*; *Schiller et al., 2008*).

Tim17 and Tim23 form the core of the complex onto which all other subunits assemble (*Dekker et al., 1997*; *Kozany et al., 2004*). Both proteins are predicted to span the inner membrane with four transmembrane (TM) helices. The TMs of Tim17 and Tim23 share limited sequence similarity; however, the two proteins cannot substitute for each other (*Ryan et al., 1994*). The first two TMs of Tim23 constitute at least part of the translocation channel of the TIM23 complex, and they actively respond to changes in the membrane potential across inner membrane (*Alder et al., 2008a*; *Alder et al., 2008b*; *Malhotra et al., 2013*). Mutations of the same segments impair the interaction of Tim23 with Tim17 and thus destabilize the entire TIM23 complex (*Demishtein-Zohary et al., 2015*). TM3 and TM4 of Tim23 appear to have a stabilizing role in the complex as their deletion is not lethal (*Pareek et al., 2013*). In contrast, essentially nothing is known about the TMs of Tim17.

## Results and discussion

### The GxxxG motifs in transmembrane segments of Tim17

We set out to investigate the roles of individual TMs of Tim17 for its function. A schematic presentation of Tim17 topology shows that this protein contains several GxxxG motifs throughout its four predicted TMs (*Figure 1A*). GxxxG motif and its more general variant with a consensus sequence (small)xxx(small) are frequently functionally important as they are often involved in interactions between TMs of membrane proteins (*Teese and Langosch, 2015*). Sequence alignment of Tim17 proteins from a number of species demonstrates that all residues of the motifs are highly evolutionary conserved (*Figure 1—figure supplement 1A*). Helical wheel projections of individual TMs show that all the glycine, alanine and serine residues of the motifs cluster on the same sides of the putative helices (*Figure 1—figure supplement 1B*).

To understand the role of GxxxG motifs in Tim17, we exchanged each conserved residue in the motif with a bulky leucine residue and analyzed the effects of the mutations in yeast. Since deletion of *TIM17* is lethal, mutants were introduced into a Tim17 shuffling strain, a *TIM17* deletion strain rescued by a wild type (wt) copy of the protein encoded on a URA plasmid. The ability of the mutant versions of Tim17 to support growth was then assessed on a medium containing 5-fluoroorotic acid (5-FOA) that selects against URA plasmid. Viable colonies were obtained with all mutants (*Figure 1B*), suggesting that a certain flexibility in packing of transmembrane segments of Tim17 can be tolerated by yeast cells.

### Effects of mutations in the GxxxG motifs

We analyzed the growth of GxxxG motif mutants. At 37°C, six mutants grew visibly slower as compared to wt cells (*Figure 2A*). Slower growth was observed irrespective of whether the cells were grown on fermentable (glucose, YPD) or non-fermentable carbon source (glycerol, YPG). The six temperature-sensitive (*ts*) mutations map to TMs 1 to 3: G25L and G29L in TM1, G62L and G66L in TM2 and G95L and G99L in TM3. G99L grew more slowly than wt even at 30°C.

We then investigated whether the slower growth of *ts* mutants is due to an impaired import competence of the TIM23 complex. When protein import into mitochondria is impaired, the precursor form of matrix-localized Hsp60 accumulates in the cytosol (*Gambill et al., 1993*). The cells were first grown at 30°C and then shifted to 37°C. Analysis of total cell extracts showed accumulation of Hsp60 precursor in all *ts* mutants (*Figure 2B*). The amount of accumulated precursor correlated with the severity of the growth phenotype. In G99L mutant, the mature form of Hsp60 was hardly detected. Impaired import via the TIM23 pathway was also observed when radiolabeled precursor proteins were imported into isolated mitochondria (*Figure 2—figure supplement 1*). The reduction of import efficiency correlated with the severity of the growth phenotype also in these experiments. The defects were observed irrespective of whether matrix translocation (*Figure 2—figure supplement 1A–C*) or lateral insertion (*Figure 2—figure supplement 1D–I*) via the TIM23 pathway was analyzed, although reduction of matrix translocation was typically more pronounced. Importantly, import of a TIM23-independent protein, ATP/ADP carrier (AAC), occurred with essentially the same efficiency in all types of mitochondria (*Figure 2—figure supplement 1J–L*), demonstrating that the mutations specifically affected import via the TIM23 pathway.

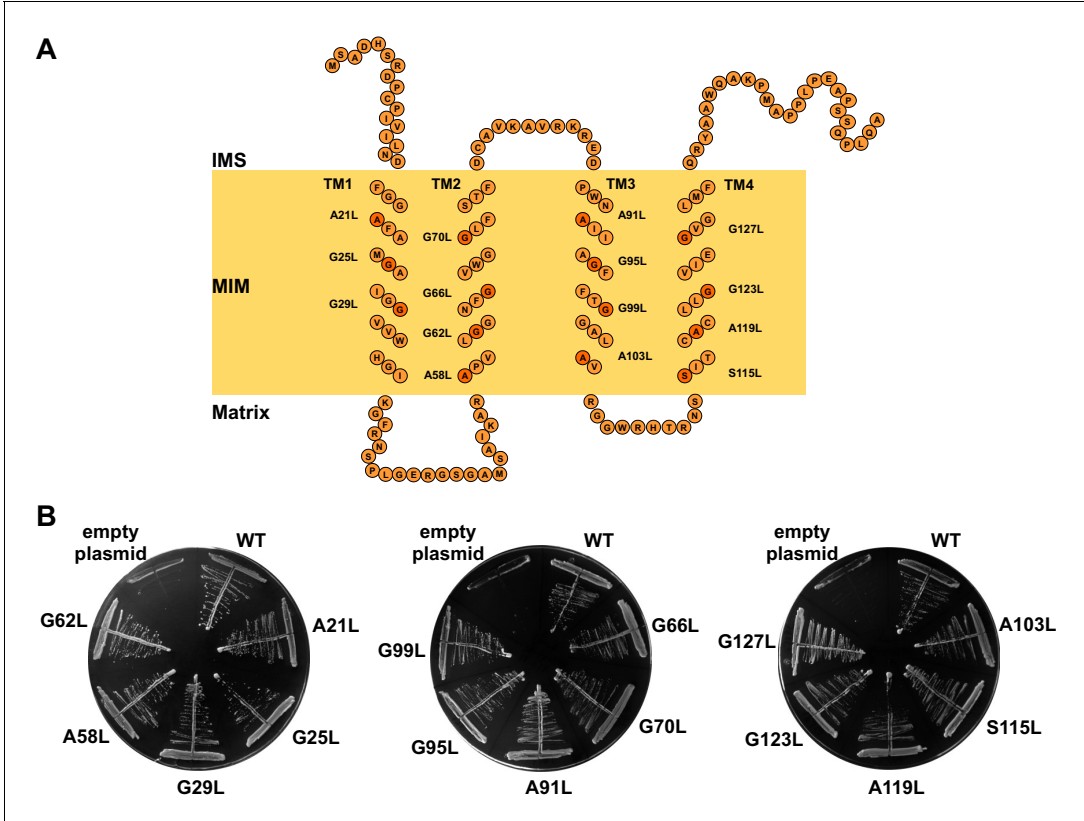

**Figure 1.** The GxxxG motifs in transmembrane segments of Tim17. (**A**) Schematic representation of predicted topology of Tim17. Glycine, alanine or serine residues in GxxxG and GxxxG-like motifs are highlighted. IMS, intermembrane space. MIM, mitochondrial inner membrane. (**B**) Each residue of the motifs was replaced by leucine. The ability of obtained mutants to complement *TIM17* deletion was analyzed on a medium containing 5-FOA. Empty plasmid and a plasmid encoding a wt version of Tim17 were used as negative and positive controls, respectively.

The following figure supplement is available for figure 1:

**Figure supplement 1.** Conserved GxxxG motifs in the transmembrane segments of Tim17.

To obtain molecular insight into the *ts* phenotypes of Tim17 mutants, we analyzed the effects of the introduced mutations on the formation of the Tim17-Tim23 core complex. To this end, wild-type cells and cells carrying *ts* forms of Tim17 were grown under permissive conditions, mitochondria were isolated, solubilized in digitonin-containing buffer and analyzed by Blue Native (BN)-PAGE. In wt mitochondria, as expected, a dominant ca. 90 kDa complex, consisting of Tim17 and Tim23, was observed (*Figure 2C*) (*Chacinska et al., 2005*; *Dekker et al., 1997*; *Popov-Celeketić et al., 2008*). In contrast, the levels of the 90 kDa complex and of the higher molecular weight species were drastically reduced in all *ts* mutants of TMs 1 and 2. SDS-PAGE analysis of the same samples (*Figure 2C*, lower panel) shows that the levels of Tim17 and several other essential TIM23 subunits are indistinguishable among different mitochondria, demonstrating that the absence of Tim17-containing complexes is not due to instability of the protein but rather due to the involvement of TMs 1 and 2 of Tim17, directly or indirectly, in the interaction with Tim23. This resembles the previous finding that TMs 1 and 2 of Tim23 are important for the interaction with Tim17 (*Alder et al., 2008b*; *Demishtein-Zohary et al., 2015*). Intriguingly, the 90 kDa complex was present at essentially wt levels in both TM3 *ts* mutants, suggesting that TM3 has only a minor role in the stability of the 90 kDa Tim17-Tim23 core complex.

The normal levels of the Tim17-Tim23 core complex in the TM3 mutants of Tim17 raised the question as to the molecular background of their *ts* phenotype. To address this question, we isolated mitochondria from yeast cells expressing His-tagged versions of the TM3 *ts* mutants and

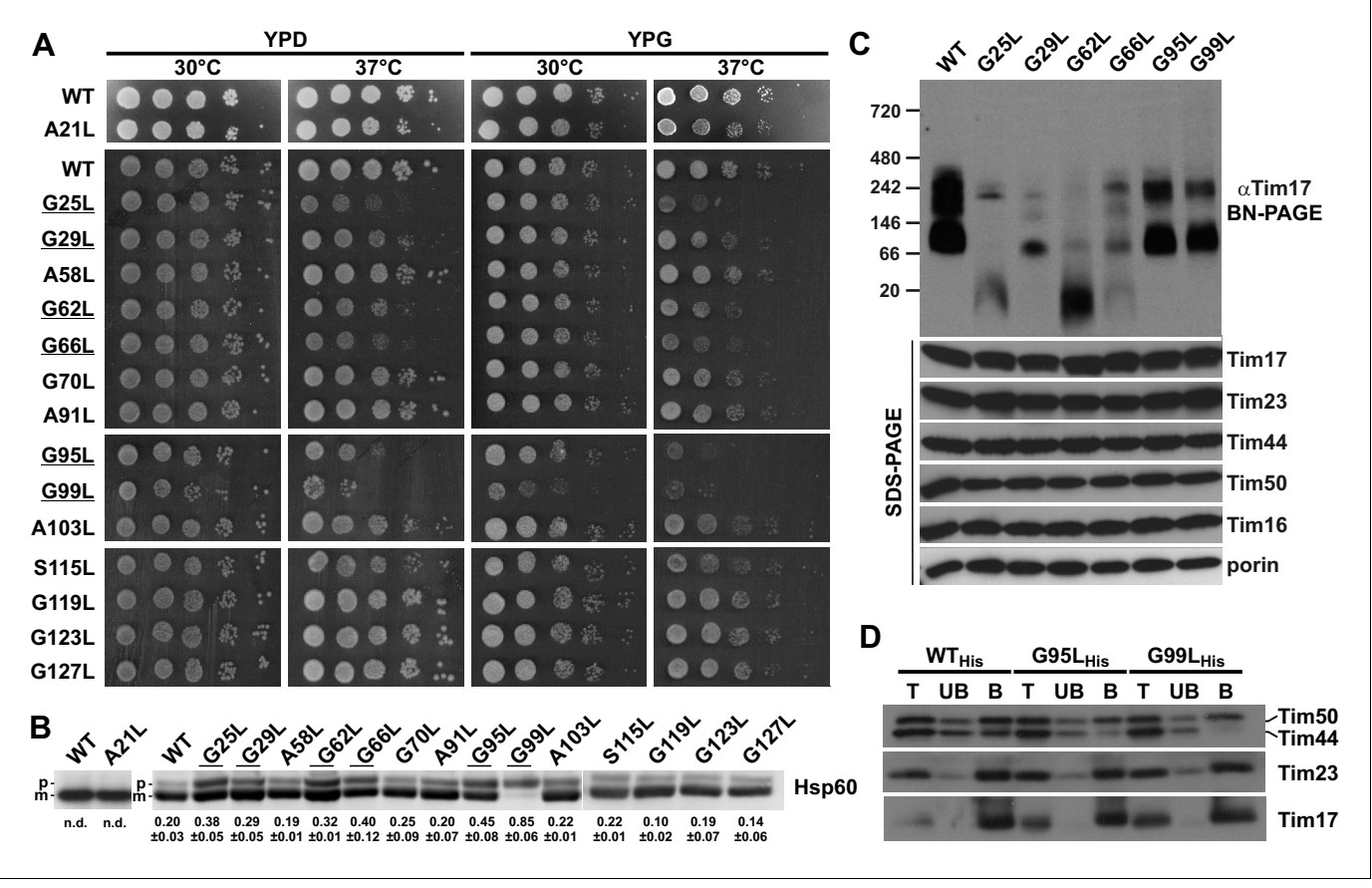

**Figure 2.** Effects of mutations in GxxG motifs. (A) Ten-fold serial dilutions of cells were spotted on YPD and YPG plates and incubated at 30°C and 37°C. Temperature-sensitive mutants are underlined. (B) Total cell extracts of cells grown at 37°C were analyzed by SDS-PAGE and immunoblotting with Hsp60 antibodies. p, precursor and m, mature forms of Hsp60. Fraction of precursor form compared to total signal (precursor plus mature form) was quantified from three experiments and shown as mean with standard deviation. n.d. – none detectable (in three experiments). Temperature-sensitive mutants are underlined. (C) Isolated mitochondria, as indicated, were solubilized with 1% digitonin and analyzed by Blue Native- (upper panel) and SDS-PAGE (lower panel) and immunoblotting with indicated antibodies. (D) Mitochondria were isolated from cells expressing indicated Tim17 versions with a C-terminal His-tag, solubilized in digitonin-containing buffer and cleared lysates were incubated with Ni-agarose beads. After three washing steps, specifically bound material was eluted with Laemmli buffer containing 300 mM imidazole. Total (T, 20%), unbound (UB, 20%) and bound (B, 100%) fractions were analyzed by SDS-PAGE and immunoblotting with indicated antibodies.

The following figure supplement is available for figure 2:

**Figure supplement 1.** Temperature-sensitive mutations in Tim17 affect in vitro import of proteins by the TIM23 complex.

performed pull down assay using Ni-agarose beads. His-tagged but otherwise wt Tim17 efficiently bound to the Ni-agarose beads, bringing down Tim23, Tim50 and Tim44 (*Figure 2D*). Similar amounts of Tim23 were bound to the Ni-agarose beads when the two TM3 mutant mitochondria were analyzed, confirming the results obtained by BN-PAGE described above. Binding of Tim50 was also indistinguishable among the three types of mitochondria, in agreement with the notion that Tim50 is recruited to the complex mainly through its interaction with the IMS-exposed domain of Tim23 (*Geissler et al., 2002*; *Gevorkyan-Airapetov et al., 2009*; *Yamamoto et al., 2002*). In contrast, binding of Tim44 to the beads was impaired in both mutants. In the G99L mutant, essentially no Tim44 was found in the bound fraction.

## Region of Tim17 involved in binding of Tim44

We sought to map the region of Tim17 involved in binding of Tim44 more precisely. Although it is likely that during translocation of proteins into mitochondria Tim44 may reach deep into the

membrane (*Kanamori et al., 1997*), it is generally accepted that the major part of Tim44 is exposed to the matrix, suggesting that its binding site should be found in more matrix-facing regions of Tim17. A binding site of Tim44 has not been found in the matrix-exposed loop between TMs 1 and 2 of Tim17 (*Ting et al., 2014*). Since our mutant analysis suggests an involvement of TM3 in the interaction with Tim44, we examined the predicted loop between TMs 3 and 4 for a potential Tim44 binding site. To this end, we first screened positions 104 to 113 by mutating the first and the second five residues in a row to alanines. The mutant of the first half of the loop resulted in a lethal phenotype, whereas the second mutant was viable (*Figure 3A*). This result suggests that the residues potentially involved in Tim44 binding are found in the region 104–108. Close inspection of this region revealed a highly conserved arginine residue at position 105 (*Figure 1—figure supplement 1A*). We replaced this arginine with an alanine residue. The R105A mutant was viable but displayed a severely reduced and temperature-sensitive growth even on fermentable carbon source and was

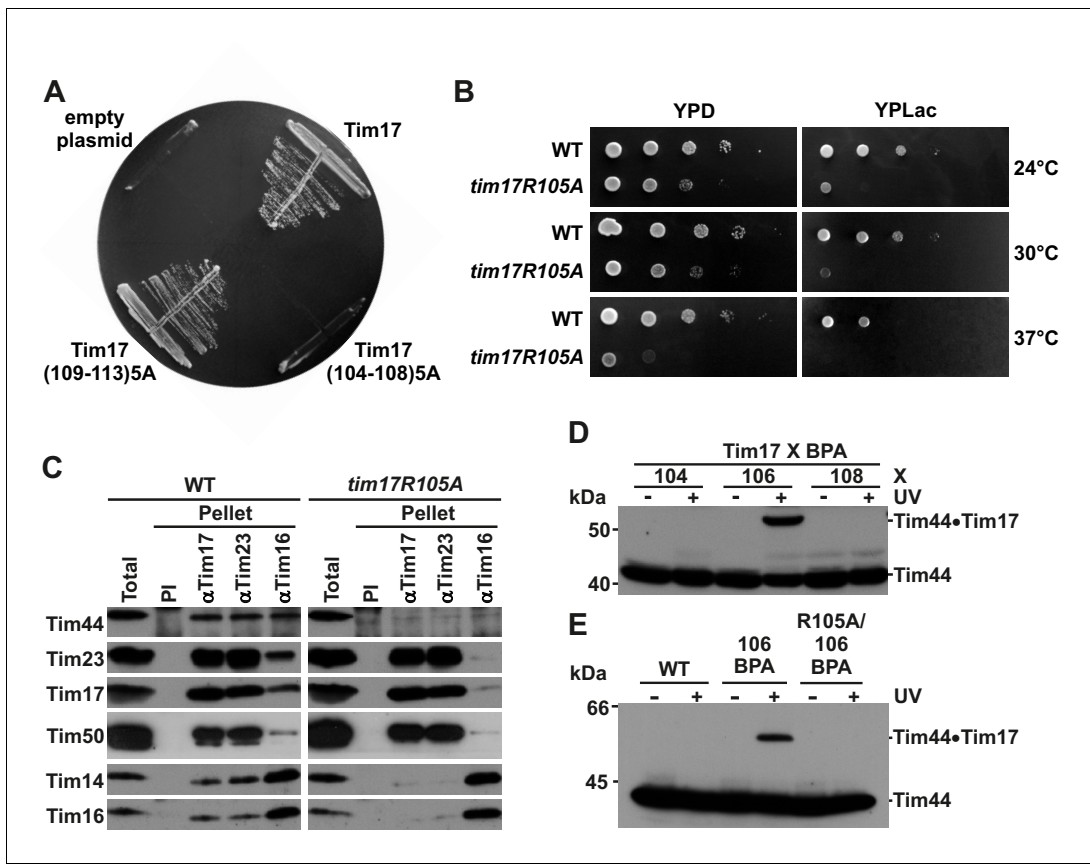

**Figure 3.** Identification of a region of Tim17 involved in binding of Tim44. (**A**) Alanine-scanning mutagenesis of Tim17 region encompassing residues 104 to 113. Five residues in a row were mutated to alanines and the ability of the mutants to complement *TIM17* deletion was analyzed on a medium containing 5-FOA. Empty plasmid served as a negative control and a plasmid encoding wt Tim17 as a positive control. (**B**) Ten-fold serial dilutions of cells were spotted on YPD and YPLac plates and incubated at indicated temperatures. (**C**) Isolated mitochondria were solubilized with digitonin and incubated with affinity purified antibodies to Tim17, Tim23 and Tim16 prebound to Protein A-Sepharose. Antibodies from preimmune serum (PI) were used as a negative control. Total (15%) and material specifically bound to the beads (100%) were analyzed by SDS-PAGE and immunoblotting with the indicated antibodies. (**D**) p-benzoylphenylalanine (BPA) was introduced at positions 104, 106 and 108 of Tim17. Where indicated, cells were UV-irradiated. Total cell extracts were prepared and analyzed by SDS-PAGE and immunoblotting with antibodies to Tim44. (**E**) Indicated cells were treated and analyzed as in panel **D**.

The following figure supplement is available for figure 3:

**Figure supplement 1.** Region of Tim17 involved in binding of Tim44.

barely viable on non-fermentable carbon source (*Figure 3B*). Submitochondrial localization of the Tim17R105A mutant was indistinguishable from wt (*Figure 3—figure supplement 1A*), suggesting that impaired biogenesis and an altered topology of Tim17 were not the reasons behind the severe growth phenotype. Impaired growth was likely due to the impaired protein import via the TIM23 pathway (*Figure 3—figure supplement 1B*). Precursor proteins translocated into the matrix or laterally inserted into the inner membrane by the TIM23 complex were imported with reduced efficiencies into mitochondria isolated from *tim17R105A* cells. Import of a TIM23-independent substrate occurred with an efficiency indistinguishable from wt. A mutant in which arginine was exchanged to lysine, thus keeping a positive charge at this position, showed a very similar growth phenotype (*Figure 3—figure supplement 1C*), suggesting that, intriguingly, the charge itself is not sufficient for function.

To analyze the effect of the R105A mutation on the assembly of the TIM23 complex, we performed co-immunoprecipitation with digitonin-solubilized mitochondria isolated from wt and R105A mutant cells (*Figure 3C*). In wt, Tim44 was immunoprecipitated with antibodies to Tim17, Tim23 and Tim16, as described previously (*Kozany et al., 2004*; *Popov-Celeketić et al., 2008*). In contrast, in the R105A mutant, the interaction of Tim44 with the TIM23 complex was severely compromised (*Figure 3C*). However, the interaction between Tim17 and Tim23 in the R105A mutant was indistinguishable from that in wt, in agreement with the results of the analysis of GxxxG motifs described above. Lack of Tim44 recruitment also led to an almost complete dissociation of Tim14-Tim16 pair from the Tim17-Tim23 core, supporting the previous finding that Tim44 is the major component that connects the Tim17-Tim23 with the Tim14-Tim16 subcomplex (*Kozany et al., 2004*). Lack of coprecipitation of Tim44 with Tim23 antibodies from lysates of the R105A mutant mitochondria further supports the notion that Tim17 is the major component for recruitment of Tim44 to the Tim17-Tim23 core (*Banerjee et al., 2015*). Essentially the same observations were made when Tim17R105K mutant mitochondria were analyzed by coimmunoprecipitation (*Figure 3—figure supplement 1D*), again suggesting that it is not the positive charge per se that is functionally and/or structurally important.

To further exclude potential secondary effects of the R105A mutation on the TIM23 complex, we attempted to obtain positive evidence for the involvement of this region of Tim17 in the interaction with Tim44. To this end, we introduced an unnatural amino acid p-benzoylphenylalanine (BPA) in vivo in several positions along this region using an orthogonal suppressor tRNA and a cognate amino acyl tRNA synthetase pair (*Kim et al., 2013*; *Shiota et al., 2015*). Upon activation with UV light, BPA crosslinks to any C-H bond in the vicinity and the obtained size shift can be detected by SDS-PAGE and immunoblotting. When BPA was introduced at position 106 of Tim17, an additional band was observed upon immunoblotting with Tim44 antibodies (*Figure 3D*). This band was present only in UV-treated cells demonstrating the specificity of the crosslink. To unambiguously confirm that this band represents a crosslink between Tim17 and Tim44, cells containing a His-tagged version of Tim17(106BPA) were UV irradiated, solubilized in SDS-containing buffer to break all noncovalent interactions and incubated with Ni-agarose beads (*Figure 3—figure supplement 1E*). The crosslinked species was efficiently recovered in the bound fraction, in contrast to the non-crosslinked Tim44. Since this band was recognized by Tim44 antibodies and it bound to Ni-agarose beads, we conclude that it represents a crosslink between Tim17 and Tim44. In the end, we made a version of Tim17 containing both the R105A mutation and BPA introduced at position 106. In this mutant, no crosslink between Tim17 and Tim44 was observed (*Figure 3E*), providing further support for the involvement of R105 in Tim44 binding.

## Conclusions

Here, we have functionally dissected the TM segments of Tim17. Our data show that the TMs of Tim17 have functionally distinct roles. This notion is further supported by the observation that in Tim17, unlike in Tim23 (*Pareek et al., 2013*), TMs 1 and 2 are not sufficient to support the function of the full length protein (*Figure 4A*). Our data support a model in which TMs 1 and 2 are involved, directly or indirectly, in the interaction with Tim23 (*Figure 4B* and *Figure 4—figure supplement 1*). Previously, TMs 1 and 2 of Tim23 were similarly implicated in the interaction with Tim17 (*Demishtein-Zohary et al., 2015*). Although the exact arrangement of the TMs remains currently unknown, the available data suggest that the first two TMs in the members of this protein family are involved in formation of higher order structures. A recently identified disulfide bond in Tim17 appears to play

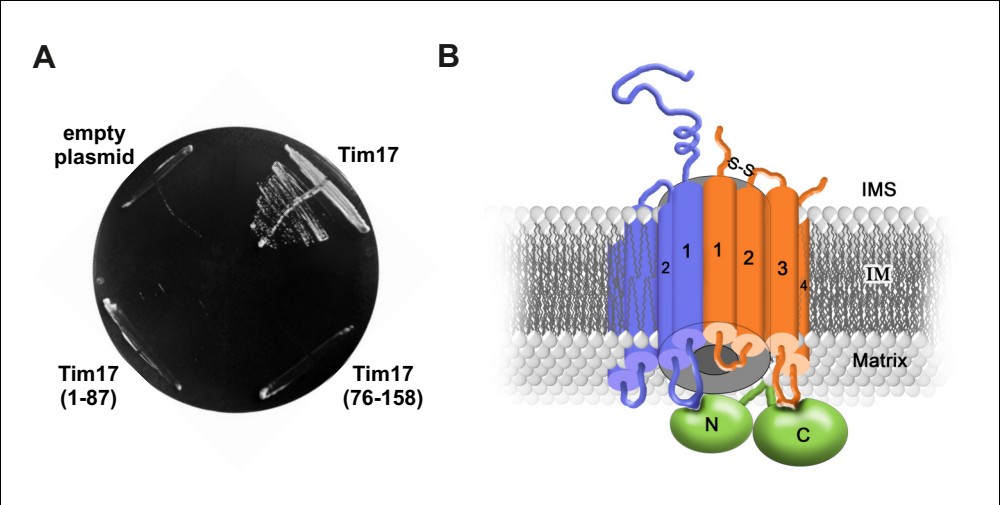

**Figure 4.** Transmembrane segments of Tim17 have functionally distinct roles. (A) The ability of the indicated mutants to complement *TIM17* deletion was analyzed on a medium containing 5-FOA. (B) A model of Tim17 (orange) and its interactions with Tim23 (blue) and Tim44 (green). See text for details.
The following figure supplement is available for figure 4:

**Figure supplement 1.** Schematic representation of Tim17 and its interactions.

a role in this process (*Mokranjac, 2016*; *Ramesh et al., 2016*; *Wrobel et al., 2016*). Whether the first two TMs of Tim17 are directly involved in channel formation, as shown for Tim23, awaits future experiments. In addition, we found that the second half of Tim17, and in particular a strictly conserved arginine residue at the matrix-facing side of IM, play an essential role in recruitment of Tim44 (*Figure 4B*). Thus, Tim17 may bridge the translocation channel and the import motor of the TIM23 complex suggesting how the translocating proteins are guided between the two.

# Materials and methods

## Site-directed mutagenesis

Site-directed mutagenesis was done according to standard molecular biology techniques. The mutations were generated on a pRS314 plasmid containing the *TIM17* ORF under the control of its endogenous promoter and 3′UTR. Where indicated, Tim17 contained a C-terminal His-tag that was introduced using standard molecular biology techniques. For in vivo site-specific crosslinking, triplets encoding residues 104, 106 and 108 were changed to *amber* stop codons using a p415GPD plasmid encoding a C-terminally His-tagged version of Tim17 as a template. All mutations were confirmed by sequencing.

## Yeast strains and media

The plasmids encoding wt and mutant versions of Tim17 were introduced into a Tim17 shuffling strain. The Tim17 shuffling strain was made by transforming haploid YPH499 wt yeast cells with a wt copy of Tim17 on a URA plasmid pVT102U and subsequently deleting the chromosomal copy of *TIM17*. The pRS314 plasmids coding for wt or mutant versions of Tim17 under the control of endogenous promoter and 3′UTR were transformed into the Tim17 shuffling strain and the ability of various mutants to support growth of yeast cells was analyzed on medium containing 5-FOA which selects against the wt copy of Tim17 on the URA plasmid.

For in vivo site-specific crosslinking, the Tim17 shuffling strain was co-transformed with the p415GPD plasmid encoding a version of Tim17 with an *amber* stop codon at a specified position and pBpa2-PGK1 + 3SUP4-tRNA$_{CUA}$ plasmid (*Chen et al., 2007*). The latter plasmid encodes for an orthogonal *amber* suppressor tRNA$_{CUA}$ and cognate aminoacyl-tRNA-synthetase specific for BPA.

Plasmid shuffling was done as described above with the difference that 5-FOA plates contained 1 mM BPA in addition. Since Tim17R105A/106BPA was not able to support growth under these conditions, this plasmid was used in the wt background. Plasmids encoding wt Tim17 and Tim17-106BPA were then used under the same conditions, to enable the direct comparison.

To test cell growth, ten-fold serial dilutions of cells were spotted on plates containing fermentable (YPD) or non-fermentable (YPG or YPLac) carbon source and incubated at indicated temperatures until colonies appeared. For isolation of mitochondria, yeast cells were grown in lactate medium containing 0.1% glucose or in YPD at 24°C. In order to be able to make direct comparisons, in all experiments wt and mutant cells were grown in parallel, in the same medium and at the same temperature.

## Hsp60 precursor accumulation

Yeast cells were grown in YPD medium at 30°C to an $OD_{600}$ of 1. The cells were then diluted to an $OD_{600}$ of 0.05 and grown at 37°C to an $OD_{600}$ of 0.8. An amount equivalent to 0.8 ODs of each strain were collected, resuspended in 100 µl ice cold lysis buffer (4% [v/v] 5 M NaOH, 0.5% [v/v] $\beta$-mercaptoethanol), mixed vigorously and incubated on ice for 30 min. Samples were neutralized by addition of 2 µl of 6 M HCl and 50 µl of 3x Laemmli buffer were added, mixed by vortexing and analyzed by SDS-PAGE and immunoblotting using antibodies against Hsp60.

## Ni-agarose binding assay

Mitochondria (300 µg) were resuspended in 500 µl SH buffer (0.6 M sorbitol, 20 mM HEPES/KOH, pH 7.4) and reisolated by centrifugation at 20,000 rcf for 10 min. They were then resuspended in 300 µl solubilization buffer (20 mM TRIS/HCl, 80 mM KCl, 1% digitonin, 20 mM imidazole, 2 mM PMSF, pH 7.4) and solubilized on an overhead shaker for 15 min at 4°C. Insoluble material was removed by centrifugation for 20 min at 20,000 rcf at 4°C. The cleared extracts were added to 25 µl Ni-agarose beads, prewashed and equilibrated with solubilization buffer containing 0.05% digitonin. Samples were incubated on an overhead shaker at 4°C for 45 min. Unbound material was collected and, after three washing steps, specifically bound material was eluted with Laemmli buffer containing 300 mM imidazole. Samples were analyzed by SDS-PAGE followed by immunoblotting.

## Blue native polyacrylamide gel electrophoresis

Mitochondria were solubilized at 2 mg/mL with 1% digitonin in 20 mM TRIS/HCl, 80 mM KCl, 10% glycerol, 2 mM PMSF, pH 8.0 for 15 min at 4°C. Insoluble material was removed by centrifugation and 2 µl of 5% Coomassie Brilliant Blue-G were added per 40 µl of supernatant. Samples were loaded on a 4–16% Native PAGE Bis-Tris Gel (Life Technologies) and ran according to manufacturer´s instructions. The gels were blotted onto a PVDF membrane and analyzed using antibodies to Tim17.

## Co-immunoprecipitation

Mitochondria were solubilized at 1 mg/mL with 1% digitonin in 20 mM TRIS/HCl, 80 mM KCl, 10% glycerol, 2 mM PMSF, pH 8.0 for 15 min at 4°C. Insoluble material was removed by centrifugation and cleared extracts incubated with Protein A-Sepharose beads with prebound antibodies against Tim17, Tim23 and Tim16. Preimmune serum served as a negative control. After 30 min incubation on an overhead shaker at 4°C, nonbound material was removed, beads were washed three times and specifically bound material eluted with Laemmli buffer. Samples were analyzed by SDS-PAGE and immunoblotting.

## In vivo site-specific crosslinking

Yeast cells expressing BPA-containing versions of Tim17 were grown in selective glucose medium supplemented with 1 mM BPA until OD of 0.8 was reached. The cultures were split into two halves and centrifuged. The cells were resuspended in 20 mM HEPES/KOH, 200 mM KCl, pH 7.4. One set of cells was UV-irradiated for 1 hr on ice and the other was kept in dark as a control. Cells were harvested, total cell extracts were prepared and analyzed by SDS-PAGE and immunoblotting.

To enrich the crosslinked products, 40 ODs of cells were collected, treated with 0.1 M NaOH, resuspended in 50 mM TRIS/HCl, pH 8.0, 200 mM NaCl, 1% SDS and boiled for 5 min at 95°C. After a clarifying spin, the supernatant was diluted with 50 mM TRIS/HCl, pH 8.0, 300 mM NaCl, 0.5%

TritonX-100, 1 mM PMSF, 20 mM imidazole and incubated with Ni-agarose beads for 30 min at 4°C. The beads were washed three times and the specifically bound proteins eluted with Laemmli buffer containing 450 mM imidizole. Samples were analyzed by SDS-PAGE and immunoblotting.

### Miscellaneous

All the experiments shown are typical representatives of experiments performed at least three times. To ensure reproducibility, all the critical experiments were not only performed with at least three biological replicates but were also repeated by at least two researchers, in some cases even in two different laboratories.

## Acknowledgements

This work was supported by the German-Israeli Foundation for Scientific Research and Development (GIF-1012/08), Israel Science Foundation (ISF-1507/13) and Deutsche Forschungsgemeinschaft (MO1944/1–1). RB is recipient of a DAAD fellowship for PhD students. We thank Petra Robisch and Zdenka Stanic for expert technical assistance, Dr Andreas Ladurner for his continuous support and Dr Peter Schultz, the Scripps Research Institute, for providing the pBpa2-PGK1 + 3SUP4-tRNA$_{CUA}$ plasmid.

## Additional information

### Funding

| Funder | Grant reference number | Author |
|---|---|---|
| German-Israeli Foundation for Scientific Research and Development | GIF- 1012/08 | Walter Neupert<br>Abdussalam Azem<br>Dejana Mokranjac |
| Israel Science Foundation | ISF-1507/13 | Abdussalam Azem |
| Deutsche Forschungsgemeinschaft | MO1944/1-1 | Dejana Mokranjac |
| Deutscher Akademischer Austauschdienst | | Rupa Banerjee |

The funders had no role in study design, data collection and interpretation, or the decision to submit the work for publication.

### Author contributions

KD-Z, Data curation, Formal analysis, Validation, Visualization, Writing—original draft, Writing—review and editing; UG, Data curation, Formal analysis, Validation, Visualization, Writing—review and editing; MM, Data curation, Formal analysis, Writing—review and editing; RB, Data curation, Funding acquisition, Validation, Visualization, Writing—review and editing; WN, Supervision, Funding acquisition, Writing—original draft, Writing—review and editing; AA, Conceptualization, Supervision, Funding acquisition, Validation, Writing—original draft, Writing—review and editing; DM, Conceptualization, Data curation, Formal analysis, Supervision, Funding acquisition, Validation, Visualization, Writing—original draft, Writing—review and editing

### Author ORCIDs

Walter Neupert, http://orcid.org/0000-0003-0571-4419
Dejana Mokranjac, http://orcid.org/0000-0002-4005-6979

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
