## [Decision Letter]

Thank you for submitting your article "Role of Tim17 in coupling the import motor to the translocation channel of the mitochondrial presequence translocase" for consideration by *eLife*. Your article has been favorably evaluated by Vivek Malhotra (Senior Editor) and three reviewers, one of whom, Klaus Pfanner, is a member of our Board of Reviewing Editors. The reviewers have opted to remain anonymous.

The reviewers have discussed the reviews with one another and the Reviewing Editor has drafted this decision to help you prepare a revised submission.

Summary:

The presequence translocase of the inner mitochondrial membrane (TIM23) is a remarkable complex and versatile molecular machine that transports several hundred different nuclear-encoded precursor proteins to their final destination in the inner membrane or matrix of mitochondria. How the different functional modules of TIM23 – receptor, channel, and motor – cooperate with each other is still under debate. Since high-resolution structural information on the organization of the TIM23 core complex is missing, thorough biochemical analysis is urgently required to better understand this central player in mitochondrial protein biogenesis.

Tim17 and Tim23 are the core subunits of the TIM23 complex. Tim23 and Tim17 have 4 transmembrane (TM) segments and are homologous to each other, yet in contrast to Tim23, the functions of Tim17 are only poorly characterized. In the present study, the authors performed mutagenesis of the conserved GxxxG motifs in TM1-4 in Tim17, and found that TM1 and TM2 are important for interactions with Tim23, and R105 in the matrix-facing loop between TM3 and TM4 plays a role in recruiting Tim44 to the TIM23 complex. The experiments were carefully and logically performed, and the obtained results are important for further understanding of the structure-function relationship of the TIM23 complex. The manuscript is concisely written, and the data are presented in a reader-friendly manner.

The findings provide the first functional dissection of the essential protein Tim17 and thus considerably improve our mechanistic understanding of protein import into mitochondria via the presequence pathway.

Essential revisions:

1) Do the distinct mutants of TIM17 generated by the authors differentially affect the import of precursor proteins into the mitochondrial matrix and inner membrane? The authors identified a number of temperature-sensitive yeast mutants. In addition to the in vivo experiments presented in the manuscript, the analysis of preprotein import into isolated mitochondria will provide important information and the authors may consider to perform a heat shock prior to or after isolation of mitochondria. The authors should use this approach to analyze at least one mutant that fails to stably interact with Tim23, one mutant in TMS3 and one TM3/4 loop mutant that fails to recruit Tim44, using a matrix-targeted preprotein and an inner membrane-sorted preprotein.

2) Figure 2 – decrease in the amount of the 90 K Tim23-Tim17 complex by mutations in TM1 and TM2 of Tim17 may not be sufficient to claim that TM1 and TM2 of Tim17 are directly involved in interactions with Tim23. The authors should either provide direct experimental results like crosslinking of BPA in TM1 and TM2 to Tim23 or they should re-write the Discussion to indicate that the current results do not fully rule out the possibility of TM1 or TM2 not directly interacting with Tim23

3) Figure 3 – the authors should use WT mitochondria prepared from cells grown in YPD as done for R105A mitochondria.

---

## [Author Response]

Essential revisions:

1) Do the distinct mutants of TIM17 generated by the authors differentially affect the import of precursor proteins into the mitochondrial matrix and inner membrane? The authors identified a number of temperature-sensitive yeast mutants. In addition to the in vivo experiments presented in the manuscript, the analysis of preprotein import into isolated mitochondria will provide important information and the authors may consider to perform a heat shock prior to or after isolation of mitochondria. The authors should use this approach to analyze at least one mutant that fails to stably interact with Tim23, one mutant in TMS3 and one TM3/4 loop mutant that fails to recruit Tim44, using a matrix-targeted preprotein and an inner membrane-sorted preprotein.

The ability of the TIM23 complex to transport proteins into two different directions, across and into the inner membrane, is one of its most intriguing characteristics, however, its molecular basis is still only poorly understood. We agree with the reviewers that in vitro import experiments into isolated mitochondria may provide important functional insight into differential sorting of proteins and we followed the suggestion to perform these experiments. We did in vitro imports of one matrix targeted preprotein and two laterally sorted preproteins, one that is motor-dependent and one that is motor-independent. In addition, we imported a TIM23-independent preprotein as a control to exclude general defects of mitochondria. To obtain a complete picture, we used isolated mitochondria of all six identified *ts* mutants in TM1-3 as well as *tim17R105A* and performed in vitro imports into mitochondria that were preincubated for 30 min at 37°C prior to import reaction or were left untreated. Overall, the in vitro import experiments recapitulated the import defects observed in vivo – the reduction of the in vitro import efficiency correlated with the severity of the growth phenotype. Matrix translocation was generally more affected than motor-independent insertion but we observed no obvious difference among the mutants in respect to differential sorting. Importantly, a TIM23-independent precursor protein was imported in all mitochondria with essentially the same efficiency. Import defects via the TIM23 pathway were visible, irrespective of whether mitochondria were exposed to heat shock or not but the exposure to heat shock made them more obvious. We therefore felt that presenting only the data obtained upon pretreatment would be sufficient, however, we can include the other results as well, depending on your decision. in vitro imports into isolated mitochondria from *ts* mutants in TM1-3 are presented in the newly added Figure 2—figure supplement 1 and the imports into *tim17R105A* are added to Figure 3—figure supplement 1. The text was also modified accordingly.

2) Figure 2 – decrease in the amount of the 90 K Tim23-Tim17 complex by mutations in TM1 and TM2 of Tim17 may not be sufficient to claim that TM1 and TM2 of Tim17 are directly involved in interactions with Tim23. The authors should either provide direct experimental results like crosslinking of BPA in TM1 and TM2 to Tim23 or they should re-write the Discussion to indicate that the current results do not fully rule out the possibility of TM1 or TM2 not directly interacting with Tim23

We completely agree with the reviewers that the data presented in the manuscript do not exclude the possibility of TM1 or TM2 not directly interacting with Tim23. However, we neither had any intention to claim this nor, we believe, we did so. We stated that TM1 and TM2 of Tim17 *are involved in the interaction* with Tim23 which, in our opinion, leaves both possibilities open. To make this point clearer we changed the text to now read “… demonstrating that the absence of Tim17-containing complexes is not due to instability of the protein but rather due to the involvement*, direct or indirect,* of TMs 1 and 2 of Tim17 in the interaction with Tim23.” In the Conclusions, the corresponding sentence now reads “Our data support a model in which TMs 1 and 2 are involved, *directly or indirectly*, in the interaction with Tim23”.

3) Figure 3 – the authors should use WT mitochondria prepared from cells grown in YPD as done for R105A mitochondria.

There is a misunderstanding and we apologize for it. In all experiments, wild type and mutant cells were always grown in parallel – in the same medium, at the same temperature and even in the same shaker. Mitochondria were then subsequently also isolated in parallel. This is our common laboratory policy introduced to be able to make direct comparisons between wild type and mutant cells and to minimize, as much as possible, any potential effects of minor variations in media composition, growth conditions and isolations of mitochondria. We modified the Materials and methods section to clarify this point.